# Effects of Additional Plyometric Training on the Jump Performance of Elite Male Handball Players: A Systematic Review

**DOI:** 10.3390/ijerph20032475

**Published:** 2023-01-30

**Authors:** Damjan Jakšić, Stefan Maričić, Nemanja Maksimović, Antonino Bianco, Damir Sekulić, Nikola Foretić, Patrik Drid

**Affiliations:** 1Faculty of Sport and Physical Education, University of Novi Sad, 21000 Novi Sad, Serbia; 2Sport and Exercise Sciences Research Unit, University of Palermo, 90133 Palermo, Italy; 3Faculty of Kinesiology, University of Split, 21000 Split, Croatia; 4High Performance Sport Center, Croatian Olympic Committee, 10000 Zagreb, Croatia

**Keywords:** plyometric, stretch-shortening cycle, jump performance, team handball

## Abstract

Handball is one of the most popular team sports around the world that has become physically very demanding, related to both competitions and daily training sessions. Optimal training programs are necessary to improve performance, especially when taking into account the frequency of jump shots (vertical jumps) and handball feints (changes of directions) during daily training. The main objective of the present study is to review the available literature systematically, and to determine what are the effects of additional plyometric training on handball players’ vertical jump abilities. According to the Preferred Reporting Items for Systematic Reviews and Meta-Analysis, six studies were selected after a systematic search through four digital databases: Web of Science, PubMed, Scopus, and ScienceDirect. The available scientific articles show that plyometric training alone or in combination with standard handball training, lasting at least six weeks, and including two training sessions per week, contributes to improving the performance of elite- or national-level handball players. Plyometric training is recommended to improve performance, as well as to maintain explosive strength parameters during the season.

## 1. Introduction

Handball is one of the most popular team sports in the world, especially in Europe, with a total of about 25 million players participating in it [1]. It represents an intermittent sport that includes many body contacts and duels between opponents and many technical and tactical elements that serve to outwit the opponent, making this sport very attractive to watch [2]. The development of the game itself has resulted in increased physical demands, related to both competitions and daily training sessions. Additionally, there is an increase in the number of high-intensity actions in a short duration in the decisive parts of a match [3,4,5,6]. Moments when important decisions are to be made strongly affect the very outcome of the match, i.e., the final score or the difference between winning and losing [7].

Team sports require good explosive strength components [8], which is manifested in almost all movements in handball, as it involves numerous movements that require the efficient recruitment of muscle fibers. According to the research conducted by Dello Iacono et al. (2015), slightly fewer than 500 high-intensity actions with almost 300 changes of direction (COD) occur during one game [9,10]. The rate of force development (RFD) is extremely important in team sports such as handball. Therefore, according to previous research [11,12,13], we consider plyometric training as very suitable for improving the RFD and overall performance. This is especially reflected in handball movements such as feints (changes of direction) and jump shots (vertical jumps) [14,15,16]. Most goals/points are scored by performing jump shots [17,18]. To develop explosive neuromuscular performance, plyometric training can be applied, which is considered a good choice since its movement structures are essentially similar to those required by a handball game (acceleration, jumping, and COD) [14,15,19,20,21,22,23]. The basis of plyometric training includes the stretch-shortening cycle (SSC), a concentric contraction preceded by a lengthening, i.e., eccentric muscle contraction [14,24]. Plyometric training appears to induce better neuromuscular coordination leading to increased power production [25,26]. Today’s training programs for developing physical abilities strive for as much sport-specificity as possible. Another reason for including additional plyometric training in the plan and program is that body mass combined with the force of gravity is sufficient for its performance [27,28], without additional loads. In these cases, the exercise load is regulated by the takeoff and landing, i.e., jump heights, and by increasing the rate of muscle stretching before a muscle contraction [29,30]. In addition to its contribution to the improvement of neuromuscular performance [22,31,32], it is also recognized as a safe form of training [21], which contributes to better injury prevention in handball players, and especially to lower extremity injury prevention [19,33,34]. Better prevention is achieved by strengthening tendons and ligaments with an overall strength and power increase [26,35,36]. For athletes, one of the main goals is to become more resistant to injuries and to reduce them to a minimum level to perform/participate in sports effectively [28]. This is especially important if we take into account previous studies that show that knee and ankle injuries are quite common in handball players, particularly in female handball players [37,38,39].

To achieve sporting success, in addition to a good vertical jumping performance and the ability to repeat vertical jumps effectively, handball players must also be able to effectively change their direction of movement several times in a short period. Plyometric training exactly consists of such movement activities, which enable efficient changes of direction and multiple jumps [40,41,42,43].

Research on the effects of plyometric training on the vertical and horizontal movements of handball players is still lacking. Therefore, we set the following objectives:
-The main objective was to review the available literature systematically and to determine if there are any, and if so, what are the effects of additional plyometric training on handball players’ vertical jump ability.-The objective was to determine whether this type of additional plyometric training improves the performance of handball players and, if so, in what way.-Finally, the authors opted for a review study that could be useful when creating training plans and programs for handball players in the future.


## 2. Materials and Methods

### 2.1. Literature Search Strategy

This paper was written and reported based on Preferred Reporting Items for Systematic Reviews and Meta-Analysis (PRISMA) guidelines and using the population, interventions, comparisons, and outcomes question model for the definition of inclusion criteria (PICO) [44], Table 1. The search process was conducted between March and May 2022. The search strategy was designed to be as broad as possible to identify all the potentially relevant literature. This systematic review included available data from the four following databases: Web of Science, PubMed, Scopus, and ScienceDirect. Additionally, articles from other sources were included as long as they were relevant to our study. The following string was applied: “Plyometric” AND “Handball,” “Stretch-shortening cycle” OR “SSC” AND “Handball” OR “Team Handball.” The screening of the articles was carried out following three phases: reading the title, reading the abstract, and reading the full text. Additionally, references of all selected articles were checked to identify other eligible studies. Journal articles from the current year were not taken into consideration. Two independent researchers decided which papers would be included in the review study. If there were disagreements, then the third researcher became involved and decided in agreement with the others. Screening processes were summarized within the flowchart PRISMA, as shown in Figure 1. 

### 2.2. Inclusion and Exclusion Criteria

To be included in the review, journal articles had to meet the following inclusion criteria: (a) journal articles must be published after 2012; (b) at least one experimental group must have performed additional plyometric training; (c) participants must be male handball players playing at the national level, (d) participants must be older than 16 years; (e) participants must have at least five years of experience in handball training or competition; (f) squat jump and countermovement jump must be tested; (g) and the studies must be original scientific studies written in English. Abstracts, conference proceedings, meta-analyses, and case studies were not considered in this review. Additionally, without articles “in press” or with early access. Only articles published in high-quality scientific journals were taken into consideration.

### 2.3. Quality Assessment

The Physiotherapy Evidence Database (PEDro) scale was used to avoid the risk of bias and to assess the methodological quality of the studies included in this review [45,46]. It has been proven to be a valid measure of the methodological quality of clinical trials [46]. The PEDro scale consists of ten conditions that can be fulfilled or not. Conditions are, respectively: random allocation, concealed allocation, baseline comparability, blind subjects, blind therapists, blind assessors, adequate follow-up, intention-to-treat analysis, between-group comparison, point estimates, and variability. A modified scale was used in this review because blinding subjects, therapists, and assessors are rarely possible in this kind of research. Items of blinding (4, 5, and 6) were removed from the scale. In the modified scale, the maximum result was 7, and the minimum score was 0.

### 2.4. Data Extraction

In this review article, the Mendeley desktop application was used to extract data from journal articles. The following data were extracted from selected studies: first author, year of publication, sample size with handball training experience, characteristics of the study population, training duration or frequency and type, testing protocols, training effects or outcomes, and PEDro score.

### 2.5. Studies Selected 

After searching the databases (Web of Science, PubMed, Scopus, and ScienceDirect), 254 studies were identified. An additional 4 records were identified through other sources. Then, duplicates were removed through Mendeley application, and 197 studies were screened with titles and abstracts reading. A total of 109 studies were excluded. After that, 82 of the studies were excluded from full-text reading. Reasons for exclusion were inappropriate sample or training type; that the experimental group did not apply additional plyometric training; that there were no appropriate comparison groups; that SJ or CMJ were not tested; and that articles were published before 2012. In the end, six studies were included in the review paper. Screening processes were summarized within the flowchart PRISMA, as shown in Figure 1. 

### 2.6. Risk of Bias Assessment or Quality Assessment

The mean score on the PEDro scale was 4.7. The highest achieved result on the scale was 6, and it was from one study that failed to fulfill only one item. Four studies did not fulfill two items, and their score was 5. One study had the lowest achieved score of 2. The study with the lowest achieved score was left into consideration. Table 2 presents the overall results of the modified PEDro scale for all studies included in the review.

## 3. Results

The main characteristics of the studies included in the systematic review are presented in Table 3. All participants in the relevant articles were male handball players, mostly playing handball at an elite level, or at least in the first and the second national leagues. A total of 180 handball players from the selected papers were included in this review article. In the selected papers, there were no injuries reported, nor were there any withdrawals. The frequency of weekly training sessions and duration of the treatment was similar in all studies. In four out of six studies, the plyometric treatment lasted eight weeks. The shortest treatment lasted five weeks, and it was only found in one study. The longest treatment lasted for the first half of the season, i.e., approximately sixteen weeks or four months. The treatment was carried out twice a week in five out of the six studies. In one study, training was conducted three times a week. Vertical jump performance was tested in every study. In every study, squat jump and countermovement jump outcomes were analyzed. Finally, the column showing training effects is presented in Table 3, which is probably the most important part of this table. Performance was improved in most tested variables, such as anthropometry, jump and sprint performance, change of direction, and maximum leg strength. Table 3 and Figure 2 below present the improvement percentages of vertical jump heights as a crucial part of this review. Five out of the six studies have confirmed the assumption that additional plyometric training can improve jumping performance. One study shows the opposite, with worsening results by about 2%. The important findings of this research indicate that additional plyometric training improved the vertical jump heights in the range of 4.5–13.9% for the squat jump and the range of 5.7–11.4% for the countermovement jump.

## 4. Discussion

The aim or purpose of this review article was to examine and determine the effects of additional plyometric training on the performance of handball players. The emphasis was on determining the effects of this type of training on explosive strength components, especially vertical jump ability expressed through the height of the jump. The jump height was our main outcome due to its availability in the found articles as well as the high correlations between short sprint performance and horizontal movements [51], which is considered very important in handball. It was tested during the squat jump (SJ) and the countermovement jump (CMJ). Five out of the six articles listed in this review study unequivocally show improvements in most of the examined abilities of handball players. The improvements are most similar regarding the aforementioned parameters of lower limb explosive strength. An exception is a study conducted by Mazurek et al. (2018) [48], where worsening results were observed in the height of the vertical jump. We assume that the pre-season treatment lasting five weeks with only 15 training sessions was of insufficient intensity to achieve improvements in the height of the vertical jump. There is an example found in basketball where significant improvements in the vertical jump height of female basketball players occurred only after six weeks of plyometric training [52]. The similarities between these activities/sports are evident. Therefore, it could be assumed that at least six weeks of plyometric treatment is required to improve the jump height in handball as well.

According to the research conducted by Čaprić et al. (2022) [53], we can see that the improvement of the neuromuscular performance of football players was achieved after a six-week treatment with additional plyometric training, and that the best results were achieved after six and eight weeks of training. Weekly, plyometric training was performed two to three times. Similar results were achieved in the studies included in this review article. An eight-week treatment including plyometric training, led to the greatest changes in the vertical jumps of the handball players [14,19,47,49]. Eight weeks of additional plyometric training resulted in the improvement percentage in SJ ranging from 9.4% to 13.9%, and in CMJ ranging from 9.1% to 11.4%. The aforementioned data have confirmed the assumption of the study conducted with football players [53]. The treatment of the longest duration, i.e., 16 weeks, referred to in this review article, achieved worse results compared to shorter treatments [50]. The results of the study conducted by Spieszny and Zubik (2018) should be regarded with skepticism, given their low score on the PEDro scale. However, it can also be observed from another perspective, as it is assumed that a longer duration of performing plyometric exercises causes weaker neuromuscular stimulation in handball players [54]. In team sports such as basketball, volleyball, and handball, where jumping is an integral part of the game technique itself, and considering previous studies, small improvements are expected due to the training stimulus, which is weaker in these athletes [54,55]. During each team training session, there is a large number of jumps performed due to the characteristics and requirements of the sport itself. The central nervous system (CNS) accumulates fatigue during a prolonged performance of the same type of movement, so the reason for worse results achieved may also be found in CNS fatigue. The explanation for the smaller percentage improvement can also be in the training history of selected handball players, as more training in the past causes less adaptation [56,57,58]. The optimal frequency and volume of plyometric training represent the main factors for the improvement of results [26].

The studies mentioned in this review article show that the best effects were achieved with two supplementary training sessions per week for eight weeks. Training with hurdle jumps and drop jumps proved to be the most effective for increasing the maximum jump height. The first half of the experimental treatment, which included hurdle jumps with a constant number of repetitions and an increase in the number of sets, achieved a better neuromuscular coordination and activation of the lower limb muscles. The other half included dropping jumps from the height of 40 cm without progressive loads [14,19]. A similar finding by Prieske et al. (2018) [59] suggests that 35 cm drop heights cause an effective utilization of the reactive strength index (RSI) for elite adolescent handball players during plyometric training. Sure, this kind of recommendation cannot be applied to all populations. High vertical ground forces (multiplied body mass) during landing is one of the factors that should be considered when determining the drop height, sets, and repetitions [60,61]. For example, athletes who weigh more than 100 kg are not recommended to jump from a height greater than 50 cm [24], because of high vertical ground forces and the great load that the musculoskeletal system suffers. According to Andrade et al. (2017) [62], for senior volleyball players achieving the best RSI values, the drop height should be 40 cm, so it could also be recommended for senior handball players.

For some detailed recommendations, trainers should first determine the structure of the team (gender, age) and their training status (e.g., level of competition). The general recommendations are to combine plyometric training with handball training (SHTP) for 8 weeks, 2–3 times a week, for 30–35 min. For players with less training experience (in handball or strength training), it is recommended to start with exercises of a smaller volume and low to moderate intensity. It is necessary to adapt and prepare the musculoskeletal system for higher loads and more demanding plyometrics. It should start with low jumps, hurdle jumps, and then continue with drop jumps in various ways.

Athletes who included additional plyometric training in addition to strength training achieved better scores in respect to the explosive power manifested through vertical jumps, ball throws, and running [63]. In this review study, the best improvement was found when plyometric training was combined with SHTP [14,19]. The reason for that probably lies in the structure of handball training, consisting of a large number of similar movement patterns and exercises. In some selected articles, the lower limb muscle volumes increased, according to the cross-sectional area (CSA) [64], so it may affect the producing muscle power [19,49]. An increase in the CSA probably leads to an increase in fast-twitch muscle fibers and better firing motor units and myosin-adenosine triphosphatase activity [20,42,65,66]. It is also used to improve athletes’ neuromuscular performance (agility, coordination) [32]. Well-designed plyometric training increases the tolerance of muscles and tendons to expected or unexpected stretching, at the same time increasing the lower limb force production [28,42,67,68]. Such improvements are especially significant for those with lower initial values of the jump height, since this type of training has proven to be very beneficial for these athletes [21]. In addition to improvements achieved in vertical jumps, improvements in the ability to involve sudden changes of direction (COD) were also found in most of the studies referred to in this review article [47,49].

Finally, the review article has some shortcomings or limitations. The first limitation is the small number of articles that fit the inclusion criteria. At a later stage, it is preferred to search for papers in other databases of scientific articles or with a smaller number of inclusion criteria. In this review article, only males aged 16 to 25 years were observed, without data about advancements in strength training in the future, the effects of this type of training should be examined on female handball players, both elite and amateur. The example of female basketball players shows that there was an improvement in the jump height results after six weeks of additional plyometric training [52], so this could also be tested on the sample of female handball players. Additionally, according to the PEDro scale suggestions, articles with a score lower than four should be excluded. This was not the case in this paper due to the small number of studies dealing with this topic.

The strength of this research lies in its practical application. By reading such papers, strength and conditioning coaches and handball coaches can find recommendations related to the frequency and volume of both plyometric training and exercises. Additional research on this topic is desirable to reveal the exact mechanisms that lead to improvements as well as to provide handball coaches and experts with recommendations and guidelines regarding the type and volume of training in several different age categories. The reason for the improvement may be found in better neuromuscular coordination, increased muscle size, or increased excitability of the stretch reflex (SSC).

## 5. Conclusions

Available scientific articles show that plyometric training alone or in combination with standard handball training, lasting at least six weeks and including two training sessions per week, contributes to improving the performance of elite- or national-level handball players. The greatest improvements occurred after eight weeks of plyometric training. The improvements were obvious in several variables, but the improvement percentage was the highest in the variables related to vertical jumps. There were changes in the maximum jump height ranging from 9.1% to 13.9%. Plyometric training is recommended to improve performance, as well as to maintain explosive strength parameters during the season.

## Figures and Tables

**Figure 1 ijerph-20-02475-f001:**
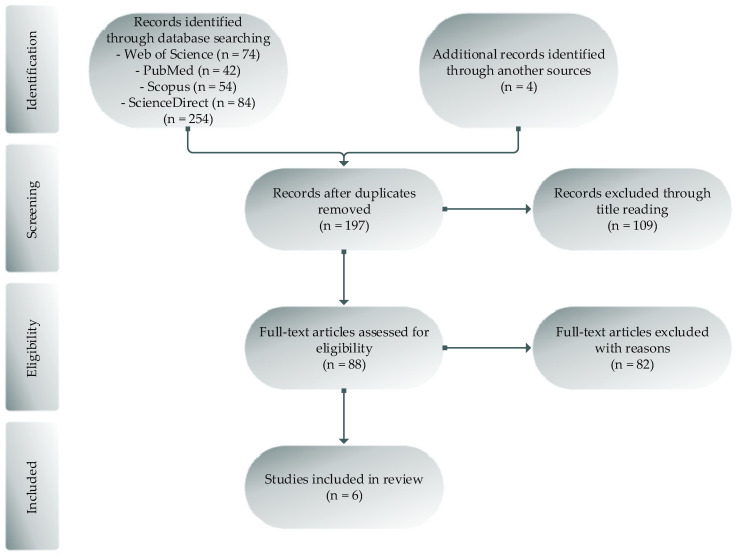
PRISMA flow chart.

**Figure 2 ijerph-20-02475-f002:**
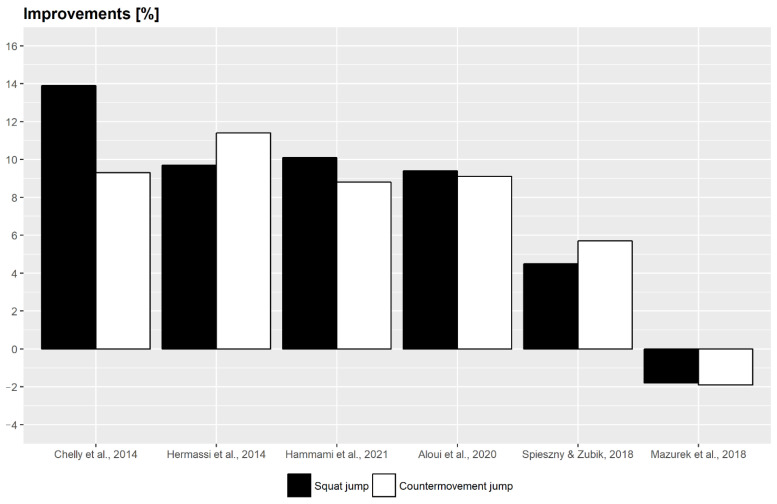
Effects of additional plyometric training on vertical jump height [14,19,47,48,49,50].

**Table 1 ijerph-20-02475-t001:** PICO question model.

PICO Components	Review Article Components
**P**opulation	Elite male or national-level handball players
**I**ntervention	Plyometric training: additional or combination
**C**omparison	Control group: without plyometric intervention
**O**utcome	Jump performance: especially jump height

**Table 2 ijerph-20-02475-t002:** Physiotherapy Evidence Database (PEDro) ratings of the included studies.

Study	1	2	3	4	5	6	7	8	9	10	Score
Chelly et al., 2014 [19]	+	-	+	x	x	x	+	+	+	+	6
Hermassi et al., 2014 [14]	+	-	+	x	x	x	-	+	+	+	5
Hammami et al., 2021 [47]	+	-	+	x	x	x	+	+	+	-	5
Mazurek et al., 2018 [48]	+	-	+	x	x	x	+	+	+	-	5
Aloui et al., 2020 [49]	+	-	+	x	x	x	-	+	+	+	5
Spieszny and Zubik, 2018 [50]	-	-	+	x	x	x	-	-	+	-	2

Notes: 1—random allocation; 2—concealed allocation; 3—baseline comparability; 4—blind subjects; 5—blind therapists; 6—blind assessors; 7—adequate follow-up; 8—intention-to-treat analysis; 9—between-group comparison; 10—point estimates and variability. (+) items fulfilled; (-) items not fulfilled; and (x) items removed.

**Table 3 ijerph-20-02475-t003:** Characteristics of included studies and training effects.

Authors	Sample and Handball Training Experience	Weekly Training Load	Experimental Training Frequency	Type of Training	Measurements/Testing Protocols	Outcomes	EG	CG
Chelly et al., 2014 [19]	23 elite male hp 17.2 ± 0.5 years EG = 12 CG = 11 HTE = 7.2 ± 1.1 years	Five training sessions lasting 90 min and one official game.	Two times per week for 2 months.	CG, handball training and strength sessions; EG, handball plus plyometric training (push-ups and hurdle and drop jumps).	- Anthropometry. - Force–velocity test for the upper and lower limbs. - Jump performance. - Handball throwing. - Sprint running.	- Thigh and leg muscle volume.	⇧*	⬄
						- Maximal anaerobic power.	⇧*	⬄
						- SJ height, power, and force.	⇧*	⬄
						- CMJ height and force.	⇧*	⬄
						- Jump velocities.	⬄	⬄
						- Thrown ball velocity.	⇧*	⬄
						- First-step velocity, 5 m velocity, and maximal velocity.	⇧*	⬄
Hermassi et al., 2014 [14]	24 elite male hp 20.03 ± 0.3 years EG = 14 CG = 10 THE = 12.4 ± 2.1 years	Six training sessions lasting 90 min and one official game.	Two times per week for 2 months.	CG, only handball training; EG, handball plus plyometric training (hurdle and drop jumps 5–10 sets × 10 repetitions).	- RSA 6 x (2 × 15 m) shuttle sprints. - Jump performance. - Leg power (7-second all-out cycling test).	- RSA_best._	⇧*	⬄
						- RSA_TT._	⇧*	⬄
						- RSA_dec_.	⇧*	⇧*
						- SJ and CMJ height.	⇧*	⇧
						- Peak and relative peak power.	⇧*	⬄
Hammami et al., 2021 [47]	32 male hp EG = 17 (16.6 ± 0.5 years) CG = 15 (16.5 ± 0.8 years) the > 5 years	Five training sessions lasting 90–120 min and one official game.	Two times per week for 2 months.	CG, standard handball training program (SHTP); EG, replaced a part of SHTP by HIIT with plyometric exercise twice per week.	- 30 m sprint test (5 m, 10 m, 20 m, and 30 m). - Modified agility *T*-test (T-half test). - Illinois modified test. - Jump performance - Repeated sprint *T*-test. - 20-meter shuttle run test.	- Sprint test (5 m, 10 m, 20 m, and 30 m).	⇧*	⬄
						- Agility test best time.	⇧*	⬄
						- Best time.	⇧*	⬄
						- SJ and CMJ height.	⇧*	⇧*
						- The best time, average time, and total time.	⇧*	⇧*
						- AMS and PMOI.	⇧*	⬄
Mazurek et al., 2018 [48]	PLYOmetric group; n = 14 Standard jump training group CG = 12 20.2 ± 2.2 years HTE = 8.4 ± 6.3 years	Eight training sessions including five conditioning, two resistance training, and three sport-specific.	Three times per week for five pre-season weeks.	- Hurdles jumps 2 × 10. - Vertical jumps 3 × 8. - Stride jumps 3 × 8. - Double leg 5 × 5. - Drop jumps 3 × 6 (20-40-60 cm). - Drop + hurdle 3 × 6 (20 cm + 76 cm; 40 cm + 76 cm; 60 cm + 76 cm).	- Anthropometry. - Repeated sprint ability. - Jumping ability. - Maximal oxygen uptake. - Aerobic power at the anaerobic threshold.	- Fat mass (%).	⇩	⇩
						- P_rsa max (W) and (W/kg)._	⇧	⇧
						- SJ and CMJ height.	⇩	⇩
						- Drop jump height.	⇧	⇩
						- VO2max.	⬄	⬄
						- PAT.	⇧	⇧
Aloui et al., 2020 [49]	CG = 15 (18.1 ± 0.5 years) EG = 14 (17.7 ± 0.3 years) HTE = 6.3 ± 0.7 years	Five training sessions lasting 90 min and one official game.	Two times per week for 8 weeks.	- EG replaced the part of technical-tactical training with plyometric training with elastic bands. - CG maintains a standard training regimen.	- Muscle volumes. - Force–velocity test (cycle-ergometer). - Sprint performance. - Repeated COD. - Jump performance. - 1-RM strength.	- Leg and thigh muscle volume.	⇧*	⇧*
						- Body fat %.	⇧	⬄
						- Absolute and relative muscle power.	⇧*	⇧
						- Sprint.	⇧*	⇧
						- Repeated COD best, average, and total time.	⇧*	⇧*
						- SJ and CMJ height.	⇧*	⇧
						- Back-half squat.	⇧*	⇧
Spieszny and Zubik, 2018 [50]	28 professional hp EG plyo = 8 (21.1 ± 2.17 years) CG = 12 (23 ± 3.05 years) EG strength = 8 (23.1 ± 2.53 years) HTE > 6 years	Five training sessions including three handball-specific and two resistance training lasting 30–45 min.	Two times per week for 4 months (15–16 weeks; the beginning of September to the end of December 2014).	EG plyo, two additional plyometric training twice a week (medicine ball throws, jumping over obstacles, hurdles, drop jumps, skipping, and multi-jumps (3–4 sets 5–10 repetitions)); EG strength, additional strength training CG maintains a standard training regimen.	- Anthropometry. - Jump performance. - 10 s trial on cycle-ergometer. - Ball-throwing velocity.	- Body weight.	⬄	⬄
						- CMJ height and maximal power.	⇧*	⇧
						- SJ height.	⇧*	⇧
						- Maximum and relative power.	⇧*	⇧*
						- Flight speed of the thrown ball.	⇧*	⇧*

Legend: EG—experimental group; CG—control group; hp—handball players; HTE—handball training or competition experience; **⇧**—increase; **⇩**—decrease; **⇧***—significant increase; **⬄**—insignificant change; SJ—squat jump; CMJ—countermovement jump; VJ—vertical jump; RSA—repeated sprint ability; RSA_best_—RSA best time; RSA_TT_—RSA total time; RSA_dec_—RSA performance decrement; P_rsa max_—maximal power in repeated sprint ability; AMS—maximum aerobic speed; PMOI—predicted maximal oxygen intake; PAT—aerobic power at the anaerobic threshold; RM—repetition maximum; COD—change of direction; SHTP—standard handball training program; HIIT—high-intensity interval training.

## Data Availability

Not applicable.

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
