# Peer review of "Effects of Additional Plyometric Training on the Jump Performance of Elite Male Handball Players: A Systematic Review"

_ijerph, 2023, doi:10.3390/ijerph20032475_

Round 1

Reviewer 1 Report

The strong point of the work is the description of the research methodology. The authors thoroughly described the selection and selection of articles that they used in their work. I notice, however, that the criteria used for the selection are not adequate/sufficient for a reliable assessment of pliometric training. One of them is the age of the participants. Athletes aged 14-16 quickly adapt to most training stimuli due to the low level of strength preparation. Hence the very large increases in jump height (13.9% SJ jump and 11.4% in CMJ Jump). In addition, the SJ jump score may be included as an indirect measure to evaluate the RSI, Strength Deficit Index, or SJ/CMJ ratio. Therefore, these parameters will be more important for evaluating SCC and plyometric training than SJ jump height. Another piece of information not included was the number of hours of handball and strength training. Players at different levels of weekly training load will react differently to the training stimulus. It is worth looking at and describing the methods used. It is worth adding a description of the methods of using plyometric training. In the found articles, the above training is used in various training methods that can clearly affect the quality of this training (e.g.: HIIT, Handball + Plyo, Plyo program). Which were the most effective? In my opinion, the discussion chapter needs to be clarified and significantly extended. It is very vague, which may be ineffective for readers who are not familiar with plyometrics, e.g. Drop Jump with a 40cm box will not be recommended for all players. The DJ must be clearly individualized to be effective (criteria for evaluating power, ground contact time, jump height) - this should be clearly emphasized. The range of repetitions and the intensity of the exercises performed are also important, which should be extracted from the articles you know. The duration of plyometric training is also debatable. On the one hand, it is true that the period of up to 6-8 weeks is the optimal time for the body to adapt to the load. On the other hand, in any type of strength training, similar trends are observed. Would the variability of loads and intensity extend the period of adaptation and progression in plyometric training? Have the authors encountered such reports in the articles they read?

To improve the quality of the work, the introduction should be focused and the discussion should be clarified and detailed. It is surprising that such a small number of articles meeting the undemanding criteria set by the authors is surprising. Therefore, despite the selection of several works, the discussion should use the remaining articles to develop the description and explanations.

Author Response

First of all, thank you for your time and such a constructive review, it means a lot to us and our further research work. We appreciate it very much. We used “Track Changes” in MS Word so you can easily see the changes.

The strong point of the work is the description of the research methodology. The authors thoroughly described the selection and selection of articles that they used in their work.

Our response: Thank you for the observation.

 I notice, however, that the criteria used for the selection are not adequate/sufficient for a reliable assessment of pliometric training. One of them is the age of the participants. Athletes aged 14-16 quickly adapt to most training stimuli due to the low level of strength preparation. Hence the very large increases in jump height (13.9% SJ jump and 11.4% in CMJ Jump).

Our response: We took that into account when selecting the articles, and that is why players from selected papers are all older than 16. Also, that is the reason why we set a requirement of at least five years of experience in handball training or competition. With that requirement, we tried to approximate the initial state of abilities of all involved handball players. In Table 2, we added training experience for every selected article, changed inclusion criteria, and set the minimum age of participants at 16.

It is a slightly more significant improvement, but it is within the range of previous research, with improvements ranging from 5 to 15% after the plyometric training program.

 In addition, the SJ jump score may be included as an indirect measure to evaluate the RSI, Strength Deficit Index, or SJ/CMJ ratio. Therefore, these parameters will be more important for evaluating SCC and plyometric training than SJ jump height.

Our response: Thank you for your observation. We agree that these parameters are better for evaluating SSC. Still, we decided on a direct measure of vertical jump height because of high correlations between sprint performance of short distances (10m and 20m) and horizontal movement, especially in the adolescent period and older, which is very important in handball. We added this in the discussion part.

Another piece of information not included was the number of hours of handball and strength training. Players at different levels of weekly training load will react differently to the training stimulus. It is worth looking at and describing the methods used. It is worth adding a description of the methods of using plyometric training.

Our response: Thank you for the advice. Sure it is worth adding. We added the frequency and duration of training sessions in a table, and now it should be much clearer to understand the weekly training load of each group of subjects. Also, we added some explanations in the discussion part.

In the found articles, the above training is used in various training methods that can clearly affect the quality of this training (e.g.: HIIT, Handball + Plyo, Plyo program). Which were the most effective?

Our response: From the selected articles, we noticed the biggest improvements when plyometric training is combined with a standard handball training program. Also, the combination of HIIT and plyometric training gives very similar improvements. The smallest improvements were observed in a combination of plyometric and strength training. We added an explanation in the discussion part.

In my opinion, the discussion chapter needs to be clarified and significantly extended. It is very vague, which may be ineffective for readers who are not familiar with plyometrics, e.g. Drop Jump with a 40cm box will not be recommended for all players. The DJ must be clearly individualized to be effective (criteria for evaluating power, ground contact time, jump height) - this should be clearly emphasized. The range of repetitions and the intensity of the exercises performed are also important, which should be extracted from the articles you know.

Our response: Thank you for the suggestion. You are right, and it should be explained more clearly. We added an explanation in the discussion part.

The duration of plyometric training is also debatable. On the one hand, it is true that the period of up to 6-8 weeks is the optimal time for the body to adapt to the load. On the other hand, in any type of strength training, similar trends are observed. Would the variability of loads and intensity extend the period of adaptation and progression in plyometric training? Have the authors encountered such reports in the articles they read?

Our response: Thank you for the suggestion. We added some explanations in the discussion.

To improve the quality of the work, the introduction should be focused and the discussion should be clarified and detailed. It is surprising that such a small number of articles meeting the undemanding criteria set by the authors is surprising. Therefore, despite the selection of several works, the discussion should use the remaining articles to develop the description and explanations.

Our response: We added some new facts in both sections – the introduction and the discussion. We did our best to find more articles that met the inclusion criteria and looked over two more databases - Scopus and ScienceDirect. With these set criteria, we were unable to find any more articles that fit the requirements.

Reviewer 2 Report

Introduction

The introduction is well structured and provides the necessary information to justify the conduct of the systematic review.

Methods

- Please specify the PICO strategy that was carried out to perform the search equation.

- Did you include any articles by snowballing? 

Discussion

- It is important when discussing adaptations to mention the type of metabolic substrate used in these actions and the type of muscle fiber. Also if the studies performed supercompensation to obtain improvements. It is important to address this in the discussion and in the discussion if applicable.

Author Response

First, we are grateful for your time and effort in reviewing our work. Second, we used “Track Changes” in MS Word so you can easily see the changes.

Introduction

The introduction is well structured and provides the necessary information to justify the conduct of the systematic review.

Our response:  Thank you for your observation.

Methods

- Please specify the PICO strategy that was carried out to perform the search equation.

Our response: Thank you for the suggestion. We put the PICO table in the method.

- Did you include any articles by snowballing?

Our response: We found some but eventually did not include them in the review article because they needed to meet the inclusion criteria.

Discussion

- It is important when discussing adaptations to mention the type of metabolic substrate used in these actions and the type of muscle fiber. Also if the studies performed supercompensation to obtain improvements. It is important to address this in the discussion and in the discussion if applicable.

Our response: Thank you for the suggestion. We included some explanations in the discussion part.

Round 2

Reviewer 1 Report

I thank the authors for responding to my comments. The authors added the "PICO question model" and extended the search scope to two databases: Scopus, and ScienceDirect. Despite this, no new publication has been added to the article. Hence, this work can be a supplement and summary of the most important information about the use of plyometric training as a supplement to handball training. The discussion has been corrected as recommended. Although, in my opinion, it was possible to go deeper into the topic using other articles. The authors presented the results of the impact of various forms and means during plyometric training on sports performance as a supplement to handball training. Key findings included a plyometric cycle duration of 6-8 weeks twice a week.

Line: 238 - remove double 58

Line 258 - remove double 63

In summary, it is worth mentioning what was written in line 216, that it is worth using hurdles and drop jumps in plyometric training.

In the discussion section - in relation to line 225-226 - it is worth adding 2 sentences about how to properly program plyometric training for handball players, what to follow? how to control?

Add limitations - about the lack of data on the advancement in strength training of players (only handball training experience).

Kind Regards

Author Response

I thank the authors for responding to my comments. The authors added the "PICO question model" and extended the search scope to two databases: Scopus, and ScienceDirect. Despite this, no new publication has been added to the article. Hence, this work can be a supplement and summary of the most important information about the use of plyometric training as a supplement to handball training. The discussion has been corrected as recommended. Although, in my opinion, it was possible to go deeper into the topic using other articles. The authors presented the results of the impact of various forms and means during plyometric training on sports performance as a supplement to handball training. Key findings included a plyometric cycle duration of 6-8 weeks twice a week.

Thank you for the second round of review. Your comments and suggestions will mean a lot to us in our future work. As in the last round, we used “Track Changes” in MS Word.

Line: 238 - remove double 58

Line 258 - remove double 63

Our response: Thank you. We accidentally duplicated the numbers. Deleted.

In summary, it is worth mentioning what was written in line 216, that it is worth using hurdles and drop jumps in plyometric training.

Our response: We added some explanations in the text.

In the discussion section - in relation to line 225-226 - it is worth adding 2 sentences about how to properly program plyometric training for handball players, what to follow? how to control?

Our response: We added some recommendations in the discussion part.

Add limitations - about the lack of data on the advancement in strength training of players (only handball training experience).

Our response: Thank you for the suggestion. We added that in the limitations part.

Reviewer 2 Report

Accept for publication

Author Response

We thank the reviewer.